# “We Were Afraid”: Mental Health Effects of the COVID-19 Pandemic in Two South African Districts

**DOI:** 10.3390/ijerph19159217

**Published:** 2022-07-28

**Authors:** Teurai Rwafa-Ponela, Jessica Price, Athini Nyatela, Sizwe Nqakala, Atiya Mosam, Agnes Erzse, Samanta Tresha Lalla-Edward, Jennifer Hove, Kathleen Kahn, Stephen Tollman, Karen Hofman, Susan Goldstein

**Affiliations:** 1SAMRC/Wits Centre for Health Economics and Decision Science—PRICELESS SA, School of Public Health, Faculty of Health Sciences, University of the Witwatersrand, Johannesburg 2193, South Africa; atiya.sph@gmail.com (A.M.); agnes.erzse@wits.ac.za (A.E.); karen.hofman@wits.ac.za (K.H.); susan.goldstein@wits.ac.za (S.G.); 2MRC/Wits Rural Public Health and Health Transitions Research Unit (Agincourt), School of Public Health, Faculty of Health Sciences, University of the Witwatersrand, Johannesburg 2193, South Africa; jessica.price@wits.ac.za (J.P.); jennifer.hove@wits.ac.za (J.H.); kathleen.kahn@wits.ac.za (K.K.); stephen.tollman@wits.ac.za (S.T.); 3Ezintsha, Faculty of Health Sciences, University of the Witwatersrand, Johannesburg 2193, South Africa; anyatela@ezintsha.org (A.N.); snqakala@ezintsha.org (S.N.); slallaedward@ezintsha.org (S.T.L.-E.)

**Keywords:** COVID-19 pandemic and lockdown, mental health and wellbeing, healthcare workers, community members, biopsychosocial

## Abstract

The impacts of pandemics are recognized to go beyond infection, physical suffering, and socio-economic disruptions. Other consequences include psychological responses. Using a mental wellbeing lens, we analyzed COVID-19-related stressors in healthcare workers (HCWs) and community members who provided and regularly accessed health services in South Africa, respectively. From February to September 2021, during the second COVID-19 wave we conducted a qualitative study in one urban and one rural district. In-depth interviews and focus group discussions were used to collect data among 43 HCWs and 51 community members purposely and conveniently selected. Most participants experienced mental health challenges regarding multiple aspects of the COVID-19 pandemic and its resulting lockdown, with a few reporting positive adjustments to change. COVID-19 impacts on mental health were consistent among both HCWs and community members in urban and rural alike. Participants’ COVID-19-induced psychological responses included anxiety and fear of the unknown, perceived risk of infection, fear of hospitalization, and fear of dying. Physical effects of the pandemic on participants included COVID-19 infection and associated symptoms, possibilities of severe illness and discomfort of using personal protective equipment. These distresses were exacerbated by social repercussions related to concerns for family wellbeing and infection stigma. Lockdown regulations also intensified anxieties about financial insecurities and social isolation. At times when common coping mechanisms such as family support were inaccessible, cultural consequences related to lack of spiritual gatherings and limited funeral rites posed additional stress on participants. In preparation for future public health emergencies, recognition needs to be given to mental health support and treatment.

## 1. Introduction

Since the onset of the COVID-19 outbreak, concerns have been raised about the possible mental health effects of the pandemic [1,2,3,4,5,6]. Concerns about depression, anxiety, post-traumatic stress syndrome and suicide follow analyses of previous pandemics such as the 1918 flu and SARS outbreaks [1,7,8,9]. During disease outbreaks, healthcare workers (HCWs), survivors of critical illness, the elderly, children, and young adults are considered especially vulnerable to pandemic-related mental health challenges [1,10,11,12], causing severe threats to public health and wellbeing. As a result, the impact of COVID-19 and lockdown regulations goes beyond the immediate impact of disease and death [13].

In March 2020, the South African government implemented a five-week “hard lockdown” (Level 5) in an attempt to prepare health services for the COVID-19 pandemic wave of infections [14]. In the early stages of the pandemic, the government and public sector were focused on saving lives and providing food and medical emergency services. The first lockdown included a total ban on alcohol sales, and complete closure of all but essential businesses and services. In the context of high levels of poverty, poor housing, and already high unemployment, this resulted in extensive job losses and hardship, especially for poor people, resulting in “the largest social and economic shock in our lifetime” [15].

In other countries, evidence emerged that COVID-19 was having mental health effects on HCWs and the general population [10,16,17]. Earlier studies conducted in Canada, the United States (US), China, and Italy specifically showed increased rates of depression and anxiety during the pandemic. Fifty percent of Canadians reported worsening mental health, and in China, 35% reported generalized anxiety while twenty percent reported depressive symptoms [2,18,19,20,21]. As the pandemic progressed, there were various stages of lockdown with varying degrees of severity, with emerging evidence of increased mental health problems in South Africa [22,23]. At the same time, health services prioritized physical aspects of COVID-19 over other illnesses [24,25,26], without any attention to the mental wellbeing of those delivering or seeking healthcare.

Many studies on COVID-19 focus on the disease or its mental health impact on HCWs. However, evidence of the pandemic’s mental health impacts and its associated control measures in the general public is limited [18,27]. Few of these studies have focused on low-and middle-income countries (LMICs), with some showing an increase in mental distress during COVID-19 in South Africa [28,29]. Prior to the pandemic in 2018, 17% of South Africans were estimated to suffer from anxiety, depression or substance-use disorders [30]. During the pandemic, about 30% of the population was depressed [28,29]. Psychological factors play an important role in behavioral actions such as adherence to COVID-19 lockdown-related measures, influencing disease spread [31]. Research on the mental health status of specific population groups is important to understand how people’s mental wellbeing was threatened during the crisis period. This paper describes the lived experiences and impact of COVID-19 on mental wellbeing among HCWs and community members who provided and regularly accessed health services, respectively, in two South African districts. This is the first study focused on the mental wellbeing of both those who deliver and seek healthcare during the pandemic.

### Theoretical Framework

Mental health is shaped to a great extent by social, economic, and physical environments [32]. The World Health Organization’s (WHO) traditional definition of health includes mental health as the ability to cope with everyday stressors [33]. We used the biopsychosocial model of mental health [34], to understand how the pandemic affected two sub-population groups in South Africa [18,35,36,37]. The model states that mental health is influenced by the interaction of three main domains: physical, social, and psychological wellbeing [34], thereby emphasizing the need for a multi-dimensional approach to understanding mental wellness [18,34,35,36,37]. The constructs were used to organize findings and thematically categorize participants’ responses to the pandemic. Regarding COVID-19, physical/biological aspects of the framework relate to the mental health consequences of having contracted the disease, and related stress reactivity such as insomnia, panic attacks, mood changes or discomfort [38,39]. Social aspects relate to effects such as social isolation, financial and food insecurity, stigma, inability to access support structures, and numerous issues such as physical infrastructure of home spaces, which restricts or makes social distancing and quarantining difficult [15,28]. The psychological aspects deal with common issues during the pandemic such as emotions, fear, anxiety, depression, or post-traumatic stress [36]. These domains interact, resulting in socio-physical, psychophysical and psychosocial functioning effects, which are important in the causality and mitigation of mental health distress during a crisis like COVID-19 (Figure 1).

## 2. Materials and Methods

### 2.1. Study Design and Settings

This was a qualitative study comprising semi-structured in-depth interviews (IDIs) and focus group discussions (FGDs) with HCWs and community members in one rural and one urban district in South Africa from February to September 2021. The study was conducted almost a year after the COVID-19 pandemic started and the initial ‘hard’ level 5 lockdown was over (see Table 1) in Johannesburg Metropolitan municipality in Gauteng province representing an urban perspective, and MRC/Wits-Agincourt Unit study area in Bushbuckridge in Mpumalanga province, offering a rural perspective. These settings were conveniently selected due to their close geographical proximity.

### 2.2. Study Population

HCWs who provided health services and community members (aged ≥24 years) who regularly accessed health services from Johannesburg and Bushbuckridge districts’ public primary healthcare (PHC) facilities and hospitals were recruited to participate. Both participant groups were fairly homogenous.

### 2.3. Sampling, Sample Size and Participant Recruitment

Purposive and convenience sampling was used to identify community members who were selected on the basis of having missed a clinic visit during the COVID-19 pandemic, socio-demographic characteristics and health services accessed. In this study, community members were defined as people or clients who regularly sought and accessed chronic services for illnesses such as hypertension, diabetes and HIV, as well as those accessing mother and child health services in the public sector. In Johannesburg, community members were recruited from health facility waiting areas. Community member participants also gave consent for their clinical records to be accessed. In Agincourt, recruitment processes included selecting a mix of participants from villages with and without a local clinic. A clinic-link system was used to identify community members who utilized healthcare services [40]. While HCWs were defined as professionals providing health services during COVID-19. HCWs were purposively and conveniently selected based on their roles and availability, including operational managers, doctors, clinical associates, nurses and community health workers from public clinics, community health centers and district hospitals. Facility-based HCWs rotated departments. The sample size in both study settings was determined when researchers deemed that data saturation had been reached.

### 2.4. Data Collection

Data were collected using semi-structured individual IDIs and FGDs, lasting an average of 30 and 60 min, respectively. Interview guides were piloted in urban Johannesburg and adapted for data collection including being modified where necessary for the rural setting (Agincourt). Notably, COVID-19 vaccination was available towards the end of the study. Therefore, vaccination data was collected among those who participated at the late stage of the study, particularly in Agincourt. Individual interviews were conducted in person or telephonically (depending on participants’ preference) and FGDs took place in person. Trained multi-lingual field staff collected data in English and six local languages: IsiZulu, IsiXhosa, SeSotho, SePedi, and SeTswana in Johannesburg, and Xitsonga in Bushbuckridge. Participants were asked questions about their experiences and perceptions of healthcare service delivery and utilization during COVID-19.

### 2.5. Data Processing and Analysis

Interviews and discussions were recorded for verbatim transcription and translated into English (where necessary). De-identified transcripts were imported into MAXQDA software (version 2020) (VERBI Software, Berlin, Germany) for data management and analysis [41]. A deductive codebook was developed and supplemented with inductive codes. A seven-member coding team was divided into pairs and a team-lead coded transcripts independently, which were then reviewed by the rest of the team during frequent meetings. Data were initially analyzed using a thematic approach guided by the interview guide. Responses to the question “describe your experiences of the lockdown”, revealed frequent references to mental health challenges faced during the COVID-19 pandemic. Hence, we looked into further analysis of the data using a mental wellbeing lens [18,35,36,37]. During the second analysis stage, the initial mental health theme was further categorized into ten sub-themes and twenty codes. The adapted mental wellbeing framework was used to group the mental health codes into the three main constructs to conceptualize factors that affected participants’ mental wellbeing during the pandemic, into physical, social, and psychological domains [18,35,36,37]. Consequences of the removal of previously utilized psychosocial coping mechanisms such as religious and cultural practices on mental wellbeing also emerged. A fourth domain, spiritual/cultural effects, is described in the findings. Direct participant quotes are used to show qualitative evidence from the research. The Consolidated Criteria for Reporting Qualitative Studies (COREQ) guideline was used to report this study [42].

## 3. Results

### 3.1. Study Participants

Interviews were conducted with 43 HCWs (13 men and 30 women) and 41 community members (8 men and 33 women) aged ≥24 years. The two FGDs were conducted with ten community members who were all adult women (Table 2).

### 3.2. Theme 1: Physical Effects

#### 3.2.1. Experiences of Consequences of Physical COVID-19 Infection

Participants expressed difficulty in coping with COVID-19 infections for either themselves or their family. Experiences of COVID-19 infection were more common among urban participants, especially HCWs. Being diagnosed with COVID-19 made participants very anxious:


*“I felt like I’m in heaven and I’m already dead. I got scared I thought of death I thought of admission to the hospital, I thought about shortness of breath I thought about all these things that we see on social media about COVID patients. I got scared.”*
(01-Urban_HCW)

Severe sickness heightened stress among participants: *“My younger brother he is staying in Gauteng, he was infected by COVID it was bad… I was crying here at home. We found out that we could not go to Gauteng province and during that time we were not allowed to travel”* (Rural-FGD). Other participants also described death from COVID-19, which increased fear: “I was really scared because of the way people were dying even in my family we had members who died from the COVID-19” (02-Urban_CM). While high death rates scared both HCWs and community members, there was a difference between the urban and rural perceptions of risk. Particularly in rural Agincourt, none of the community member participants reported being infected, with a few being aware of someone infected with COVID-19.

For some participants, constant feelings of anxiety led to insomnia and experiences similar to panic attacks:


*“But it was almost like an anxiety attack because once I think of my family and them losing me, I would start having shortness of breath… Yes, physically I had symptoms [COVID-19] but once I start stressing, looking at the TV seeing people dying then I would all of a sudden have shortness of breath, I would breathe heavily even when I’m sleeping.”*
(01-Urban_HCW)

#### 3.2.2. Experiences of Physical Discomfort Due to Personal Protective Equipment (PPE) Use

Community members also described physical discomfort of using PPE, such as the frequent wearing of facemasks and sanitizing especially at the beginning of COVID-19: *“For me it was very difficult in such a way when I was wearing a mask my heart was beating faster than normal, like I was going to die”* (Rural-FGD). Some HCWs also expressed similar discomfort: *“Even us as staff it was difficult for us because we must sanitize everywhere we touch… [And] must wear mask the whole day”* (05-Rural_HCW). Another participant described this difficulty of PPE use, feeling as if she could not breathe:


*“Nurses were confronting me and telling me to also cover my nose... I was telling them I am suffocating, and they would tell that it is not allowed to remove mask when you are surrounded by people.”*
(Rural-FGD)

### 3.3. Theme 2: Social and Environmental Effects

#### 3.3.1. Experiences of Financial Distress

For most community members, financial stress increased during lockdown. This was due to job losses, and sources of income depleting. Experiences of financial stress within households also affected children, as caregivers could not provide adequate financial support. Some community members described how they lived in constant fear of losing their employment. Conversely, HCWs as part of essential service providers continued to receive their full benefits *“… Financially I do not want to lie it did not affect us because at work we were still paid our full salaries”* (05-Urban_HCW). Despite being financially stable, some HCWs and community members described having to support families struggling due to the pandemic. Significant financial strain resulted in distress, as described by this participant:


*“For my wife it affected her emotionally because she was not able to assist me in the house. And for our children, it affected them psychological because they were now confused that their mother is not working anymore, and daddy cannot buy them the things that they used to have.”*
(07-Urban_CM)

Many participants reported taking strain and described feelings of failing the families who were faced with food insecurities: *“It affected us badly because I could not send them [parents] money… They were telling me on the phone that they were starving, and that really affected me emotionally because I was eating here at work, but my family is starving”* (09-Urban_CM). However, for other participants lockdown enabled new access to the governments’ financial assistance through the COVID 19 social grant: *“For some of us who are not working this grant [USD25] is helping us a lot”* (Rural-FGD).

#### 3.3.2. Family Concerns and Social Isolation

HCWs were perceived to be at greater risk of COVID-19 than the general community due to their occupation: “*My family was not feeling good at all; because when they call me from Gauteng they were saying ‘take care of yourself. Then I started to understand that they are not happy because corona virus it is too scary”* (09-Rural_HCW). This shows that family members were worried about relatives who were HCWs. For other HCWs, fear of infecting family made home-life challenging. This HCW described her experience of this issue:


*“He was so scared in a way that he thought I am going to kill him [partner]. Then I told him that there’s no other way I must come back here [home] after work… He was and kept saying you must sleep with a mask on, and I could not.”*
(07-Urban_HCW)

To avoid additional sources of stress on families, some HCWs opted to withhold information about COVID-19 infection. For example, this HCW described not disclosing their confirmed infection status: *“So, that I don’t stress people with COVID, I kept it to myself, my family and colleagues”* (05-Rural_HCW). Fear of infecting family with COVID-19 resulted in some HCWs stopping social visits completely, which caused guilt and anxiety:


*“I stopped seeing people who are close to me [including boyfriend]… Because we’re exposed to such high risk and I think that that’s what I was worried about, giving it to other people. I mean if someone dies because of COVID and I had given it to them, how do you live with yourself?”*
(09-Urban_HCW)

HCWs experienced social isolation through their own will and through family members forcing it on them. Participants, who regarded family and social relationships as critical for support, described being socially isolated as emotionally difficult. In addition, there was a sense of helplessness due to the loss of independence brought about by isolation/quarantine. Confinement led to feelings of loneliness among those infected with COVID-19: *“I am sitting alone in my room, and they bring things for me and put them on my door. I cannot go out and do those things for myself like I am used to doing that was another thing that was stressing me”* (05-Urban_HCW). Some HCWs felt frustrated by the limited support they could give their children doing homeschooling: “*In terms of the system, it assumed that each and every parent is at home, so that was the most difficult thing that my kids dropped in terms of their academics. Their performance dropped a lot because I could not support them at all”* (16-Urban_HCW).

#### 3.3.3. COVID-19 Related Stigma

The issue of COVID-19-related stigma came up frequently. This was related to experiences and perceptions of how participants felt others perceived them and/or how they perceived others regarding COVID-19. For example, some participants reported being afraid to cough or sneeze in public, as they might have been labeled as “infected”, expressed by this participant:


*“It was difficult, very difficult because we were afraid of each other, it was not easy. Even now it is not easy we are unable to work freely, because you do not know where other people are coming from and what they have.”*
(21-Rural_CM)

Some HCWs described how family and community perceived them as a source of COVID-19 infection. This resulted in fears of being near and/or refusing HCWs entry into households: *“The patient said no, no, don’t come into my house because you’re going to bring me COVID from the clinic. So, patients are scared of healthcare workers*” (09-Urban_HCW). Speaking about the same matter, a rural HCW described how some community members mocked and called her names: *“Now that I am always wearing a mask, they think I have COVID-19... Now they do not even want to talk with me… Everyone was calling me by names like they were making fool of me”* (03-Rural_HCW). However, rural FGD participants argued that their perceptions of HCWs did not change, explaining that: *“We did not fear them; we just take them like normal people just like us”* (Rural-FGD). HCWs with both urban and rural pandemic experiences explained that people in the rural area were more welcoming to HCWs compared to urban, as one doctor put it:


*“It is like they [rural] value what we do, but in Johannesburg it is different. Everyone would avoid you, saying oh you are a doctor. They will start to put their masks on.”*
(17-Rural_HCW)

On the other hand, HCWs talked about some COVID-19 stigma within health facilities, particularly COVID-19 suspected patients: *“Stigma, even here at work, when our COVID first case come out we ran away they said this is a patient suspect… If it is you and want help then everybody is running away from you how you would feel?”* (07-Urban_HCW). HCWs also reported stigmatizing each other when a colleague had confirmed COVID-19:


*“When they [colleagues] see you, they look at you like you still got COVID and they look at you like you going to infect them, there was stigma among us the healthcare workers.”*
(05-Urban_HCW)

### 3.4. Theme 3: Spiritual and Cultural Effects

#### 3.4.1. Religious Gatherings and Church Attendance

The impacts of COVID-19 on religious practices such as church attendance was a widely discussed psychosocial challenge among participants in both urban and rural settings. The inability to attend church for some community members was a desideratum. Other community members said that whilst they understood the reasons for restrictions on social gatherings, it still made life difficult, as discussed:


*“If we don’t go to church, we don’t have peace... They say we must not go to church, and if a person is sick, we were going to that person and pray for him to get better, so what about now? We saw that we are going to die… But, if we can pray God will have mercy on us.”*
(Rural-FGD)

#### 3.4.2. Funeral and Cultural Practices

COVID-19 lockdown restrictions on funeral gatherings were perceived as a major challenge for many. Failure to attend funerals led to feelings of deep sadness and loss. While inter-provincial travel restrictions made provision for people to travel for funerals, people who died during COVID-19 were buried immediately. This was often limiting for those who needed to get time off work or gather money for funeral attendance. While some participants reported the inability to attend funerals, one HCW who had to bury his during the pandemic father had this to say about the rapid process:


*“According to the regulations... When someone passes on, you need to immediately come with your undertaker so you can move them to the mortuary. We were also told that we couldn’t keep him there longer than 72 h…So, we had to rush everything.”*
(04-Urban_HCW)

Rushed pandemic funerals left mourners questioning the sufficiency of their loved one’s departure and burial process. The inability to hold funerals in their cultural capacity was another issue mentioned. In the South African setting, funerals provide an opportunity to perform traditional practices, especially for elderly deceased persons. For example, funeral restrictions limited the ability for practices like body viewing:


*“It was a sad funeral, because her coffin was wrapped with plastic, we could not see her and she did [not] even go inside the house, we could not even practice the cultural things we normally do. And as she was a traditional healer, we could not do thing proper, because the undertakers have changed the way they bury people.”*
(02-Urban_CM)

The inability to grieve properly led to mental distress and guilt, as highlighted by this participant: *“Now what makes me sick is that I could not go and bury the father who gave birth to me”* (Urban-FGD). For those participants who were able to attend a funeral, some reported standing remotely during burial to avoid COVID-19 infection: *“But we were standing far from the grave because if we stood near, we could get infected… That was our biggest fear”* (Rural-FGD).

### 3.5. Theme 4: Psychological Effects

#### 3.5.1. Initial Panic and Fear of the Unknown

Fear of the unknown was described as the worst at the beginning of the pandemic. This initial pandemic panic was common among all participants, as an HCW described, *“I think the most difficult thing for myself was the fear of the unknown that you do not know if you will wake up tomorrow”* (02-Urban_HCW). Limited disease knowledge made HCWs acutely fearful: *“Initially it was difficult, we were scared at the beginning working with COVID”* (16-Rural_HCW). People were not sure about “*What was happening or would happen, and how we should act towards it. So, we were just scared*” (10-Urban_HCW). Community members also found the unknown COVID-19 frightening: “*During lockdown we were not feeling well emotionally… We were seeing it for the first time, so we were not feeling safe at all”* (21-Rural_CM). Initial uncertainty about the pandemic often led participants to ascribe other illnesses to COVID-19, heightening anxiety. Others expressed coping difficulties during this period, with some seeking professional psychological support:


*“It makes it very detrimental like this job, we are doing, [and] we are sacrificing ourselves and sacrificing our lives for this job. It is very hard I think we are not coping. I have not been coping. I even started seeking help, I am seeing a psychologist just to help me to cope with what is going on and the patients that keep on dying.”*
(17-Rural_HCW)

In trying to make sense of their experiences, some participants compared COVID-19 to previous pandemics, most notably HIV. Many believed that COVID-19 was scarier than HIV: *“This disease was terrifying because you can see the pandemic and we had to wear masks, it was scary, not like HIV. HIV was better, this was worse”* (05-Rural_HCW). The changing pandemic and its continuation through different waves also exacerbated fear.

#### 3.5.2. Fear of Infection and Infecting Others

The fear of contracting COVID-19 was almost universal. HCWs described high infection rates amongst colleagues, often reporting severe symptoms and even death, resulting in being afraid of falling ill on duty and dying: *“Until our staff started getting sick… that is when I felt like “I may die on duty”* (06-Rural_HCW). Participants indicated they constantly felt unsafe, especially HCWs:


*“I was in a constant state of fear, worried that I was not doing enough to protect myself, doubting how safe the mask is and how useful washing hands actually is? I was sometimes thinking of wearing more than one mask, just to feel more secure.”*
(04-Urban_HCW)

HCWs’ feared occupational contraction of COVID-19, *“We had stress. Especially us HCWs because we were still on duty, so we were living in a constant state of fear. Because we were the most exposed to COVID”* (05-Urban_HCW). An HCW who interacted with patients daily, screening for COVID-19 symptoms at the gate, said: *“Honestly speaking, it made me scared because now everyone is coming to the clinic and from different places… You do not know if they do have COVID. Now I had to screen all of them, so that scared me”* (01-Rural_HCW). Possible disease transmission among colleagues was described by HCWs, leaving staff feeling vulnerable and conflicted about continuing work:


*“We were not told who it was [contracted COVID-19] and I keep on asking myself who was it? What if it is the person I go with everyday outside to the field. And every team member that I came across, would ask myself is it this person?”*
(05-Urban_HCW)

Community members also expressed fear of contracting COVID-19: *“The biggest fear was to go and visit our relatives… Even if a person says he has got some flu, we were afraid that maybe it is COVID”* (Rural-FGD). Fear of contracting COVID-19 increased when the perceived risk of dying or severe illness was present, particularly in elderly persons and people living with chronic conditions: *“I would tell them that I am afraid of getting the virus they must respect that as I am a person living with the chronic condition and anytime it can infect me. I am now 54 years old and there is high blood, asthma, things like that”* (02-Urban_CM).

For many participants, fear of death was interlinked with anxieties around the implications of their deaths for their families such as breadwinners leaving behind families facing great financial strain, leaving orphans or their entire families being wiped out by the pandemic:


*“I am HIV positive, and I heard that people who are HIV positive are more susceptible to getting it. So that is why I was so scared, that if I get it, I will die and leave my children alone, or I will infect them, and we all die.”*
(07-Urban_CM)

Participants also identified situations they perceived to pose higher risks of infection. These included crowded places such as clinics, shopping malls and public transport: *“In public transport, you are not sure about your safety because there are a lot of different people”* (21-Rural_CM). Some participants reported public minibus taxis breaking COVID-19 protocols such as the number of people occupying a vehicle. Furthermore, fear of infection often resulted in clinic avoidance: “*Even when people are unwell, they fear going to the clinic, because of COVID. They would rather quarantine and get medication from home”* (01-Urban_CM).

Fear of infection included concerns about the safety and welfare of children. An HCW who was infected with COVID-19 said, *“I got scared for my kids… because I’m scared of infecting them and my kids might die”* (01-Urban_HCW). HCWs described extensive safety measures taken to limit risk, including immediate bathing, undressing at designated areas (often outside the home) and immediate washing of clothes, or keeping them separate. Another HCW described how this made them feel heightened stress:


*“There is a lot of obsession with this [pandemic]. When we get home, we take it [clothes] off, so we have to first get some sanitizer. When you enter the house, you turn straight to the bathroom to have a bath before [anything else]… It wasn’t fun because I feel like someone who is losing their mind… So I was always nervous.”*
(02-Urban_HCW)

#### 3.5.3. Fears Related to Hospitalization

HCWs and community members feared hospital admission alike: *“I never wish to get hospitalized in a COVID ward. Yeah, I was not comfortable about that. My fear was not getting infected with the virus but getting admitted”* (06-Rural_HCW). Fear of admission was described by another urban-based HCW who self-medicated as prophylaxis to prevent hospitalization:


*“I am not asthmatic, [but] at that time I thought let me rather use Asthavent [asthma medication] because if I have to get admitted, who will come visit me? I can’t go to the hospitals because in hospitals they don’t even have home remedies I’m going to die in that oxygen, no.”*
(01-Urban_HCW)

Many participants were anxious, as many individuals died once admitted to the hospital. Other hospital-related fears amongst participants included resource shortages and unpredictable COVID-19 prognosis that left people feeling anxious about their lives: *“Because of this COVID thing and because of how new it is, nobody understood it because it’s very dynamic, one minute it’s this and in one minute it’s that. So that was my main worry”* (06-Urban_HCW).

Fear of admission was compounded by hospitals’ no-visiting policies: *“I was afraid of going to the hospital and getting admitted because if you get admitted then there are no visitors allowed to see you”* (06-Urban_CM). This regulation caused distress for those who wanted to visit admitted relatives: *“We were unable to visit our relatives at hospital. It was not easy; we were hurting while we are at home. Even if you can go to hospital but you would not get in or see your relatives”* (Rural-FGD). Other participants were concerned that canceled visitations meant the inability to bid farewell to dying relatives.

Some HCWs expressed emotional exhaustion from witnessing patient deaths: *“It’s tiring to see people dying every day and you are emotionally stressed but you must keep going because they are now short staffed, and they need you”* (07-Urban_HCW). Fear of dying was intensified by negative information about COVID-19 on various mass media platforms: *“After they diagnosed me with COVID… I think that social media made me more anxious, actually, because of those things that I was reading on social media”* (11-Urban_HCW).

#### 3.5.4. Disruptions to Coping Mechanisms and Adaption to the ‘New’

Participants cited usual types of coping skills used pre-COVID-19 regulations including personal relationships, church attendance and substance use. Disruptions to previously available psychosocial support were common, while the need for continued support was left unmet both in the urban and rural setting. Lack of access to habitual coping mechanisms stressed some participants. One HCW described elements of depression: *“It affected me because now I was stressing a lot, when you are used to getting out, spending time with people, going to church, and now you are kept isolated, on your own… So I would think a lot”* (01-Urban_HCW). In their accounts of events surrounding the removal of coping strategies, some participants talked about the implications of alcohol bans and their inability to unwind after a stressful workday: *“We work with people, so we are under a lot of stress and sometimes you just want to go home and have one glass just to relax, so we didn’t have alcohol just to unwind and all that”* (10-Urban_HCW). For others, social networks were cited as a previously relied on source to de-stress:


*“We don’t get any debriefing sessions here at work, so the only debriefing that I get is visiting friends or meeting friends and going drinking, so for me it was a lot. It was actually overwhelming with the burden of saying there is this COVID thing, the stress at work doesn’t stop so it was really bad.”*
(01-Urban_HCW)

Our data showed some participants were predisposed to better stress handling. While COVID-19-induced fears and anxieties were almost universal among all participant groups, in both the urban and rural settings, some participants indicated that they adjusted well to new routines: *“It was something that came out of nowhere, we thought that this was something small and that it would last just three months, but until now we are still under lockdown, so we had to adjust to the system”* (12-Urban_HCW). This included finding social isolation as a time for rumination. Some specific coping mechanisms that seemed most effective were the use of technology such as mobile phones which became an important means of connecting with and monitoring the wellbeing of social networks, as exemplified in this quote:


*“We were unable to visit them because it was not allowed due to COVID 19 protocols. We were calling them, chatting with them, checking on how they are doing, it was very difficult for us.”*
(14-Rural_HCW)

A few participants described not experiencing COVID-19-related anxiety. For some HCWs, testing patients for COVID-19 was not a fearful exercise due to feelings of altruism and being fearless: “*My problem is that I enjoy my work”. I enjoy testing people [for] COVID… I am not scared; even now, I am not scared* (01-Rural_HCW). With time, participants became less anxious about COVID-19 and its possible consequences. Some HCWs and community members reported becoming at ease through acquiring more knowledge about COVID-19, specifically, HCWs’ confidence in diagnosis improved, and understanding the precautions required. The study commenced before the availability of COVID-19 vaccines, therefore, vaccine availability made some feel more relaxed and hopeful:


*“Honestly, I feel like the fear has subsided a little bit, unlike during the early days, especially with testimonies of different people who have survived what they have done to survive and how they survived.”*
(18-Rural_HCW)

## 4. Discussion

Since the onset of the pandemic, by May 2022 about 100,000 people had died of COVID-19 in South Africa. Deaths were highest in Gauteng with 21,000 and Mpumalanga had one of the lowest rates at 5000 [43]. The COVID-19 pandemic presented significant challenges to the mental wellbeing of a sample of HCWs and community members in urban and rural areas. COVID-19 impacts on mental health were consistent among both participant groups and across both settings. Mental distress was experienced because of both physical COVID-19 infections and because of lockdown measures implemented to reduce the disease spread. Fears and anxieties related to the pandemic included perceived risk of infection, possibilities of severe illness, financial insecurity, hospitalization, and death. Fears were exacerbated by high degrees of uncertainty and negative reporting by the media. Most participants were anxious about not only their own wellbeing, but also that of family, and were concerned about what it would mean to their families if they died prematurely. In addition to mounting COVID-19-related stressors, lockdown regulations restricted access to normal coping mechanisms generated by the inability to visit family and places of worship. Findings demonstrate the overlapping and inter-relational complex nature of mental health challenges in both participant groups and across settings during COVID-19 (Figure 2). Our data also highlights the importance of the cultural/spiritual effects not emphasized in previous mental health research on disease pandemics.

Our findings speak to existing mental health and wellbeing frameworks [18,34,35,36,37] and literature on psychological reactions during the COVID-19 crisis [1,2,3,4,5,6,7,8,9,10,11,12]. Mental health is a result of multiple negative and positive factors, which have a cumulative impact on individuals and sub-populations [34]. While feelings of fear and anxiety are a normal human response to threats such as COVID-19, however, prolonged mental distress can undermine coping abilities during and after a crisis [44,45]. In the UK, potential mitigating processes such as psychological flexibility were found to enable adaptive COVID-19 coping and accounting for differences in mental health for both acute and longer-term pandemic challenges, in keeping with findings from other work [46]. While there is a need to recognize public health mental needs during disease outbreaks [47], resource-constrained settings such as most LMICs may face difficulties in allocating resources for these services. Low-cost interventions such as telephonic counseling services have been shown as both effective at reducing mental illness associated with acute stresses such as those posed by the pandemic, as well as improving household stability and reducing the costs associated with inpatient care required should mental distress progress [48]. Furthermore, virtual consulting services are able to fill gaps created by strict adherence to COVID-19 protocols such as that which occurred in China [49]. In South Africa, support services were established to aid HCWs experiencing stress and anxiety following the COVID-19 pandemic. These included providing information around mental health, accessing counseling services and online support groups, including telephonic mental health services, SMS hotlines and training [50,51]. However, these interventions omitted the general public.

The locality of disease prevalence during a pandemic is an important factor to consider in crisis management for better mental health outcomes. The City of Johannesburg was an epicenter of most COVID-19 waves in terms of numbers [52]. This phenomenon may help explain our findings on the physical effects of COVID-19 infections and deaths that appeared more common among urban than rural participants. During future epidemics, better mental health support together with reducing psycho-physical symptoms should be a priority in formulating public mental health interventions that improve risk among both HCWs and vulnerable localities [53]. Uncertain COVID-19 incubation periods and possible asymptomatic transmission exacerbated pandemic fear and anxiety [47]. In China, fear of contracting COVID-19 and its potential consequences was also extended to worrying about family members [53]. Particular situations such as crowded health facilities were understood to pose a higher COVID-19 infection risk. This sometimes led to avoidance behavior resulting in missed clinic visits and a preference for non-crowded spaces, in keeping with experiences elsewhere [54].

Other studies have identified specific sub-populations to be vulnerable to deteriorating mental health during COVID-19 [1,10,11,12]. Individuals who may respond more negatively to COVID-19 stressors include children and personnel assisting with outbreak containment like as HCWs [55]. In addition, those who experienced financial stressors due to loss of employment during the pandemic suffered significant mental health impacts in South Africa [56]. Financial stressors were a significant burden for community members. In contrast, HCWs were mostly spared from financial stress due to ongoing employment; however, they experienced increased anxiety about the risk of death in patients and reported increased workloads, changes to work routines and adjustments to new COVID-19 protocols often leading to burnout. Similar experiences have been noted globally [57]. Stressful and traumatic events such as this impact the mental wellbeing of those who need to deliver health services effectively, thereby posing challenges to the quality of services.

From our data, HCWs suffered from poorer mental health than community members. In addition, HCW fear and avoidance were reported as critical under-recognized forms of stigmatization in the US and Canada [58]. Stigma and fear can negatively impede social and organizational outbreak control [59]. In this study, both HCWs and community members experienced some form of stigma. Although this was more frequently reported in the urban setting. HCWs also incurred unforeseen burdens following changes to schooling policies including virtual schooling, or split class schedules that resulted in children being at home despite the need for HCWs to attend work in person. Furthermore, public school closures meant a loss of access to school feeding schemes—an important intervention that addresses food insecurity for vulnerable children [60]. We assume that this may also have contributed to additional psychosocial stressors among financially struggling community members.

When facing distress, people’s health and healing is also influenced by social and spiritual factors [61]. Spiritual and cultural effects although critical have largely gone unrecognized in COVID-19-related mental health research. Religious and cultural struggles are important to acknowledge as they shape the health of individuals and sub-populations differently [62,63]. Funerals are of importance to one’s ability to process the death of a loved one, including performing proper rites and ceremonies for the ‘next life’. Globally, as part of COVID-19 control measures, governments imposed funeral restrictions on mourning periods, attendance numbers, and prohibition of culture-based practices performed before, during, and after a funeral [64]. Our data showed that these deviations from normal cultural practices caused great distress to bereaving families when they were most vulnerable. Another feature of South African funerals that was prohibited was ‘after tears’ ceremonies—a celebration of the life of the deceased in a party-like atmosphere. This can serve as an important coping mechanism after burial for mourners [65].

Notably, the absence of coping mechanisms and resilience factors that could have mitigated COVID-19 stressors under normal circumstances also played a significant role in mental wellbeing. When compared to similar previous outbreaks such as influenza, SARS, and Ebola, a review revealed that health systems failed to extrapolate learnings from these previous outbreaks to effectively respond to the public mental health aspects of COVID-19 [66]. The government felt they needed to address physical issues related to COVID-19, mental issues were not specifically considered. Evidence from this study indicates that future public health responses to pandemics should prioritize mental health and psychosocial consequences together with physical effects to effectively circumvent crises such as COVID-19 [45,67]. Specific up-to-date and accurate health information is also required, from reliable sources to lower pandemic-related psychological impacts [53]. Furthermore, public health responses need to target pandemics of excessive exposure to mass media in certain cases. For example, fake news created panic amongst the general public [55,68,69]. Public psychological crisis interventions should be formally integrated into public health emergency preparedness and response plans [47].

### 4.1. Recommendations

This analysis shows that there is a need for government to prioritize investment into mental health support programmes—within both the formal health sector and community structures. This should include training of more mental health practitioners, particularly community psychologists as there is an already existing shortage in South Africa. Results suggest a broader approach that incorporates policies to provide more and better food and financial security including strict enforcement to reduce risk factors for severe common mental disorders. Consideration should be given to enabling safer religious and cultural practices such as keeping the minimum duration of gatherings; while developing resilience strategies for coping with change such as physical, social, spiritual, and psychological support services.

### 4.2. Strengths and Limitations of the Study

A strength of this study was that it provides lived experiences and perceptions of South African residents’ mental health challenges under the COVID-19 pandemic from an urban and rural context, shedding vivid images vis-à-vis raw number data of most quantitative studies on the topic. In addition to HCWs, this analysis considers the mental health effects of the pandemic on the general public with a specific focus on those who access healthcare services at the PHC level. A limitation that could have affected the quality of our findings is possible inadequate recall from participants regarding questions about COVID-19 and lockdown experiences that occurred post the initial level 5 ‘hard’ lockdown phase (12 months prior). Furthermore, the interview guide was not designed to examine mental health per se, mental wellbeing emerged as an important and dominant theme, despite not being the original focus of the research. Another possible limitation could be the overrepresentation of women in the study. However, this is quite common in mental health studies.

## 5. Conclusions

This analysis shows that COVID-19-induced mental health outcomes were more widely experienced than previously understood, among HCWs and community members who had limited contact with people who had COVID-19. While the disease itself drove significant anxiety and fear, measures imposed by the government to curb the pandemic led to substantial financial and social strain and were equally important in contributing to poor mental wellbeing. These biopsychosocial responses to the pandemic were near universal in both urban and rural settings. In future pandemics, to improve public health and mental wellbeing among vulnerable groups, there is a need for a holistic approach to public health emergency and response that goes beyond physical disease aspects and must consider impacts on social, spiritual/cultural, and psychological wellbeing.

## Figures and Tables

**Figure 1 ijerph-19-09217-f001:**
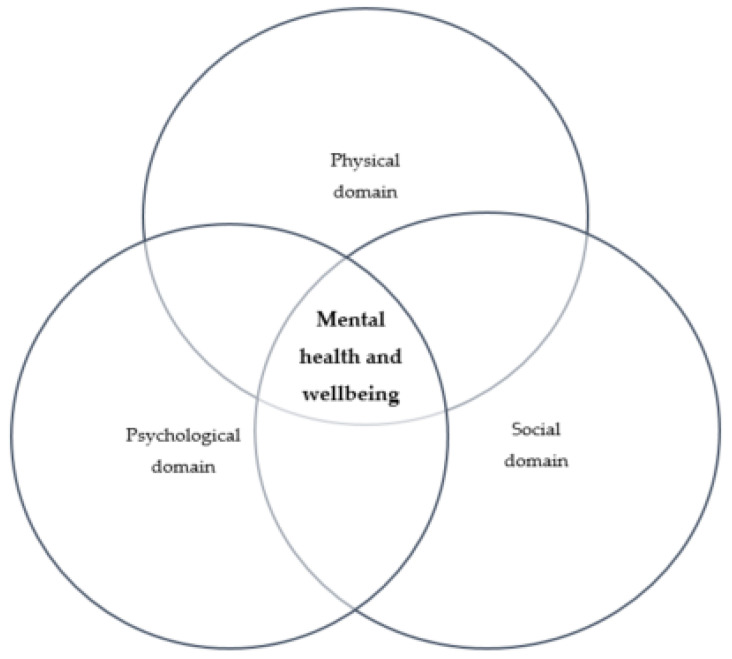
Three-dimensional mental wellbeing framework adapted for the COVID-19 crisis. Adapted from [34].

**Figure 2 ijerph-19-09217-f002:**
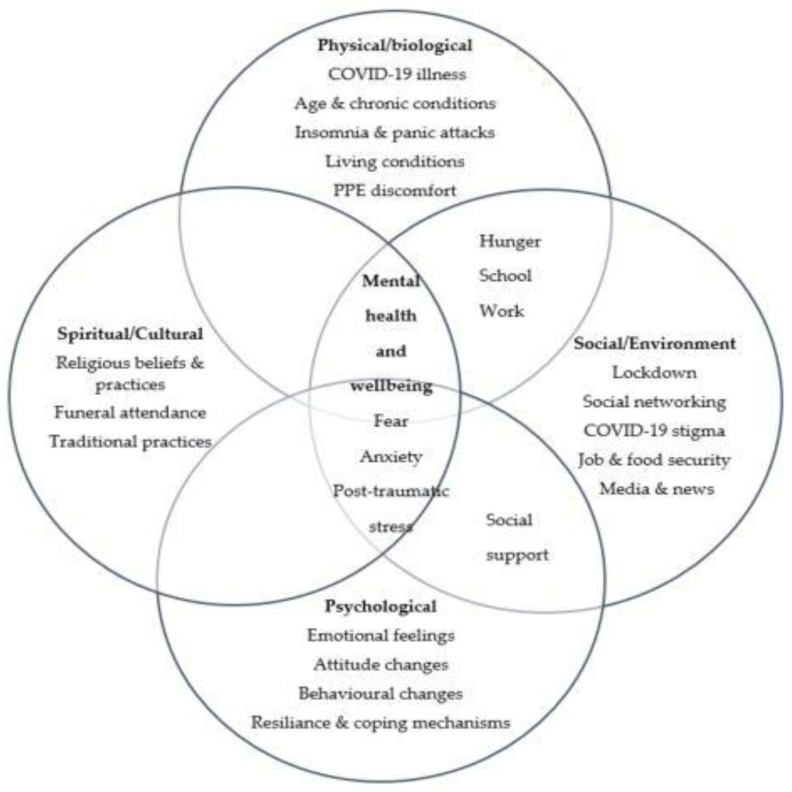
The intersection of mental health challenges among participants during the COVID-19 pandemic and lockdown in South Africa.

**Table 1 ijerph-19-09217-t001:** Summary of COVID-19 lockdown alert levels in South Africa.

Alert Level	Lockdown Objective
Level 5	High COVID-19 spread, with low health system readiness and drastic measures in place
Level 4	Moderate-high COVID-19 spread, with low-moderate health system readiness and extreme precautions
Level 3	Moderate COVID-19 spread, with moderate health system readiness and restrictions on many activities
Level 2	Moderate COVID-19 spread, with high health system readiness and restrictions on social activities
Level 1	Low COVID-19 spread, with high health system readiness and most normal activity in place

**Table 2 ijerph-19-09217-t002:** Number of interviews conducted and participants’ demographics and composition.

Participant Group and Demographics	Characteristics	Setting		
		Urban	Rural	Total
		Johannesburg	Agincourt	
Healthcare Workers (HCWs)	IDIs	25	18	43
Gender	Female	19	11	30
	Male	6	7	13
Occupation	Operational Manager	0	1	1
	Doctors	3	4	7
	Clinical Associate	1	0	1
	Nurses	15	9	24
	Community Health Workers	4	4	8
	Other	2	0	2
Facility type	Clinic	16	12	28
	Community Health Centre	9	2	11
	District Hospital	0	4	4
Comorbidities and other COVID-19 risk factors	Yes	6	5	11
	No	1	1	2
	Smoking	1	0	1
	Unknown	17	12	29
COVID-19 facing role	Yes	18	5	23
	No	6	1	7
	Unknown	2	11	13
COVID-19 infection	Positive test	7	2	9
	Unknown	18	16	34
Community Members (CMs)	IDIs	17	24	41
Gender	Female	14	19	33
	Male	3	5	8
* Service accessed	Chronic (Diabetes)	3	4	7
	Chronic (Hypertension)	6	9	15
	Chronic (HIV)	4	14	18
	Maternal Health	7	1	8
	Child Health	8	5	13
	Other	4	3	7
Community Members (CMs)	FGDs	1	1	2
Gender	Female	6	4	10
	Male	0	0	0
* Service accessed	Chronic (Diabetes)	2	0	2
	Chronic (Hypertension)	0	4	4
	Chronic (HIV)	0	0	0
	Maternal Health	2	0	2
	Child Health	2	0	2
	Other	0	0	0
Total participants		48	46	94

* CMs may have accessed more than one health service and/had more than one chronic condition.

## Data Availability

Data is available from authors upon reasonable request.

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
