# Peer review of "“We Were Afraid”: Mental Health Effects of the COVID-19 Pandemic in Two South African Districts"

_ijerph, 2022, doi:10.3390/ijerph19159217_

Round 1
Reviewer 1 Report
Manuscript focuses on an important topic and are relevant to the subject of specific issue. Method and discussion are well written with declaring its strength and weaknesses. I have no further comments on topic.
Author Response
The authors would like to thank the reviewer for the thoughtful comments.
Reviewer 2 Report
Dear Authors,
Thank you for the opportunity to review this high-quality article.
Most of the studies on the subject are quntitative (including my own) and it is very interesting to see such vivid images of what is usually presented as raw number data or information reduced to 'increased anxiety' or 'fear for family's health'.
The whole paper is meticulously prepared and clearly presented.
I see no major issues. One possible limitation could be the significant overrepresentation of women in the study. This is quite common in most studies on mental health, but could use acknowledging.
For the whole picture, it would be interesting for me (and perhaps for other readers as well) to see some of the responses of people that were not that significantly affected by the pandemic. Perhaps, you could evaluate what predisposes to better stress handling or identify some specific coping mechanisms that seemed most effective.
These are just suggestions, the article is fit for publishing as it is.
Author Response
Authors would like to thank the reviewer for their thoughtful comments. The authors agree with the reviewer in that the study is unique in its qualitative methods among other studies on this topic. However, as opposed to reducing data we believe that our study more than anything expands raw numbers and complements these from an often unheard perspective. To this end we have modified the strengths and limitations section of the paper, as follows: We rephrased a sentence on lines 680-683, pg. 18 to reads as: “A strength of this study was that it provides lived experiences and perceptions of South African residents’ mental health challenges under the COVID-19 pandemic from an urban and rural context, shedding vivid images vis-à-vis raw number data of most quantitative studies on the topic. The authors thank the reviewer for raising the point about the overrepresentation of women. We have added this to the limitation section, lines 690-692, pg. 18: “Another possible limitation could be the overrepresentation of women in the study. However, this is quite common in mental health studies.” We would like to acknowledge the reviewer’s comment, and direct them to the Results section, where we have a sub-section titled 3.4.4 Disruptions to coping mechanisms and adaption to the ‘new’, lines 477-519, pg. 14-15. Here we show findings of how some participants reported positive adjustments to change during the pandemic. This included altruism and being fearless for some HCWs, finding social isolation as a time for rumination, increased reliance and importance of mobile phones and similar technologies as means of connecting with and monitoring the wellbeing of social networks. Furthermore, as time passed and more information about COVID-19 and vaccines became available both HCWs and CMs reported to be more at ease with living with the virus. |
Reviewer 3 Report
First and foremost, I wish to congratulate the authors for shedding light on such an important topic in a timely fashion. The study aims to shed light on South African residents’ mental health challenges under the pandemic. Please find my comments below and respond to them sufficiently, as I believe answering these concerns could help the authors further enhance their work, and in turn, the readers better appreciate the study.
There is a considerable disconnect between the study findings discussed in the abstract section and the main text. Also, the title and the study are not in line with one another: the first claims that the study sets out to investigate “Unseen mental health effects of the COVID-19 pandemic in two South African districts”, while the other was stated as “Using a mental wellbeing lens, we analysed COVID-19 related stressors in healthcare workers (HCWs) and community members who regularly accessed health services in South Africa”. Please address these issues.
Could the authors please define “healthcare workers” and “community members”? These definitions should help clarify why these subpopulations are mutually exclusive, as opposed to otherwise.
On a related note, could the authors please provide a table that can help the readers better understand the study participants? Currently little is known about their medical seniority (for the healthcare professional population), educational background, presence of chronic diseases, or other COVID-19 risk factors (e.g., obesity, smoking, ability to conduct work remotely, etc.), COVID-19 infection/vaccination history, etc.
Could the authors please shed more light on the “one rural and one urban district” selected for sampling? To what extent is this selection process “purposeful” in relation to the research aim of the study?
Much information is missing in Figure 2, in part because the colour schemes and the legend hues adopted neither match nor are well-explained. Please address this issue.
Also, legends are needed to shed light on the information mentioned in Figure 3, as well as what various colour schemes mean if they bear unique meanings in relation to the said information. This is particularly relevant as both the number of factors and the colour used in Figure 1 and Figure 3 are incongruent.
The discussion section could be further improved—right now arguments are not presented in an optimally structured manner. For instance, in terms of contextualizing the research findings and practical insights this study could provide, maybe the authors could discuss South Africa first, and then other regions or the rest of the world in general?
Some minor points. Line 13, what do the authors mean by “using a mental wellbeing lens”? Overall, please consider reviewing the manuscript in detail to ensure there are no incongruences throughout.
Author Response
We would like to thank the reviewer for their insightful comments that have helped strengthen the paper. To address the perceived disconnect, we have expanded on the results section of the Abstract (now 254 words) so it speaks more to the study findings. See lines 19-32, on pg. 1: “Most participants experienced mental health challenges regarding multiple aspects of the COVID-19 pandemic and its resulting lockdown, with a few reporting positive adjustments to change. COVID-19 impacts on mental health were consistent among both HCWs and community members in urban and rural alike. Participants’ COVID-19 induced psychological responses included anxiety and fear of the unknown, perceived risk of infection, fear of hospitalization, and fear of dying. Physical effects of the pandemic on participants included COVID-19 infection and associated symptoms, possibilities of severe illness and discomfort of using personal protective equipment. These distresses were exacerbated by social repercussions related to concerns for family wellbeing and infection stigma. Lockdown regulations also intensified anxieties about financial insecurities and social isolation. At times when common coping mechanisms like family support were inaccessible, cultural consequences related to lack of spiritual gatherings and limited funeral rites posed additional stress on participants.” The Authors agree with the Reviewer’s observation of the potential disconnect and point of confusion. This steams from the fact that the primary study objective was not focusing on mental health per se; hence the original use of “unseen”. To avoid confusion, we have: 1) Deleted the word “Unseen” from the title, lines 2-3, pg. 1. It now reads as:
“We were afraid”: Mental health effects of the COVID-19 pandemic in two South African districts 2) We added a new sentence to the study limitations section on lines 688-690, pg. 18 that reads as follows: “Furthermore, the interview guide was not designed to examine mental health per se, mental wellbeing emerged as an important and dominant theme, despite not being the original focus of the research.” In light of the modifications 1) and 2) the authors strongly believe that the phrase ‘using a mental wellbeing lens’, is necessary to keep as it is at the heart of our methods. It describes how we used the mental wellbeing theoretical framework as a structure or a data mining lens to make sense of the research study data. Under Materials and Methods section, sub-section 2.3 Sampling, sample size and participant recruitment, we have re-phrased some sentences to add definitions of the study sub-populations, as per reviewer’s suggestions. This includes adding the phrases “HCWs who provided health services and community members who regularly accessed health services” where appropriate in the manuscript. The definition of the study sub-populations are as follows: Lines 143-146, pg. 5: “In this study, community members were defined as people or clients who regularly sought and accessed chronic services for illnesses such as hypertension, diabetes and HIV, as well as those accessing mother and child health services in the public sector.” Lines 151-155, pg. 5: “While HCWs were defined as professionals providing health services during COVID-19. HCWs were purposively and conveniently selected based on their roles and availability, including operational managers, doctors, clinical associates, nurses and community health workers from public clinics, community health centres and district hospitals. Facility-based HCWs rotated departments.” Table 1 had been updated to include the detailed information suggested by the reviewer (see Table 1), lines 198-200 on pg. 6-7. Health data among HCWs collected for presence of comorbidities and if the HCW’s role was COVID-19 facing. In addition, a positive COVID-19 test was mentioned by 9 of the HCWs. Some information was voluntarily provided like smoking status by 1 of the HCW participants. Information about COVID-19 vaccination status not collected among participants. Vaccinations were not yet available at the onset of this study, information about vaccination was added and collected among a few participants who participated at the end of the study, in particular the rural setting; which collected data last. We have added a sentence that specifies that study sites were conveniently selected for the purpose of having an urban and rural setting, as these areas were in close geographical proximity for the research team. Lines 117-118, pg. 4: “These settings were conveniently selected due to their close geographical proximity.” The initial Figure 2 has been replaced with another image, which better illustrates the levels of lockdown that occurred in South Africa, and the time during which the study was conducted added. More information has been added under the key to contextualize the meaning of each of the five levels. A legend has been added to Figure 3, line 576-577, pg. 16 as follows: “Figure 3. The intersection of mental health challenges among participants during the COVID-19 pandemic and lockdown in South Africa.” We have removed the colour schemes on both Figure 1 and Figure 3 to avoid the incongruences highlighted by the reviewer. The colours were to make the figures more legible and had no unique meanings. Both Figure 1, lines 102-107, pg.3 and Figure 3, line 553-577, pg. 16 have been reformatted to be in black and white. The authors thank the reviewer for the suggestion on possible restructuring of the discussion. We acknowledge that several feasible and logical structures could be utilized, nevertheless we do believe that the current presentation of an “inverted funnel” works well for our study. It is a common structure in the literature to start the discussion with a narrower and more focused summary of main results, then gradual expansion to comparisons of the results with other studies and the interpretation in the wider context of the study topic. Specifically, this paper’s Discussion section is organized according to the four domains of theoretical framework used in the study, including coping; which is critical to address when discussing mental health challenges. First, a summary of the results. Second, psychological impacts during disease outbreaks. Third, physical effects. Forth, social, economic and environmental effects. Fifth, spiritual and cultural effects. Lastly, coping and resilience, information and bringing evidence together. The phrase ‘using a mental wellbeing lens’, means that we used the mental wellbeing theoretical framework as a structure or a data mining lens to make sense of the research study data. |
Reviewer 4 Report
In the introduction section (lines 69-71) it is mentioned that “This is the first study focused on the mental wellbeing both those who deliver and seek healthcare during the pandemic.”. From the way the results are presented it is difficult to draw comparative conclusions regarding the urban vs rural environment or the two groups considered.
The authors (simply) stated that they had obtained data from 43+51=94 HCW and community members. which is a little small. This may require a brief discussion / limitation in the discussion section. (The essence of investigation is to infer the characteristics of the population from the characteristics of the sample information.)
Author Response
We thank the reviewer for the important observation. We have added a sentence in the Discussion section’s first paragraph that discusses the summary of the main findings, lines 525-526, on pg. 15 that reads: “COVID-19 impacts on mental health were consistent among both participant groups and across both settings.” In addition, we have added a new sentence in the Methods section, under sub-section 2.2 Study Population on lines 137-138 pg. 5 that reads: “Both participant groups were fairly homogenous.” In response to the comment regarding sample size, the authors would like to indicate that in qualitative studies, recruitment of a smaller number of participants is justifiable, as they are not meant to be representative of the overall population. Qualitative sample sizes are considered adequate when they satisfy two main parameters: being large enough to enhance rich insights, and small enough for an in-depth inquiry of the phenomenon under study. In addition, we collected qualitative information using focus group discussions in each setting, to ascertain if we had missed any information during the one-on-one in-depth interviews; and the data obtained was quite similar. |
Round 2
Reviewer 3 Report
I wish to congratulate the authors for their timely work. My recommendation is "accept".
Reviewer 4 Report
Accept.